# Simultaneously Improved Pulmonary and Cardiovascular Autonomic Function and Short-Term Functional Outcomes in Patients with Parkinson’s Disease after Respiratory Muscle Training

**DOI:** 10.3390/jcm9020316

**Published:** 2020-01-22

**Authors:** Chih-Cheng Huang, Yun-Ru Lai, Fu-An Wu, Nai-Ying Kuo, Yuh-Chyn Tsai, Ben-Chung Cheng, Nai-Wen Tsai, Cheng-Hsien Lu

**Affiliations:** 1Neurology, Kaohsiung Chang Gung Memorial Hospital, Chang Gung University College of Medicine, Kaohsiung 83301, Taiwan; hjc2828@gmail.com (C.-C.H.); yunrulai@cgmh.org.tw (Y.-R.L.); tsainw@yahoo.com.tw (N.-W.T.); 2Department of Biological Science, National Sun Yat-Sen University, Kaohsiung 80424, Taiwan; benzmcl@gmail.com or; 3Respiratory Therapy, Kaohsiung Chang Gung Memorial Hospital, Chang Gung University College of Medicine, Kaohsiung 83301, Taiwan; helenyoli93@hotmail.com (F.-A.W.); ellenkuo@cgmh.org.tw (N.-Y.K.); jane2793@cgmh.org.tw (Y.-C.T.); 4Internal Medicine, Kaohsiung Chang Gung Memorial Hospital, Chang Gung University College of Medicine, Kaohsiung 83301, Taiwan; 5Center for Shockwave Medicine and Tissue Engineering, Kaohsiung Chang Gung Memorial Hospital, Chang Gung University College of Medicine, Kaohsiung 83301, Taiwan; 6Department of Neurology, Xiamen Chang Gung Memorial Hospital, Xiamen, Fujian 361126, China

**Keywords:** respiratory muscle training, pulmonary function, cardiovascular autonomic function, functional outcomes, Parkinson’s disease

## Abstract

Both pulmonary function and autonomic function are impaired in patients with Parkinson’s diseases (PD). This study tested the hypothesis that respiratory muscle training (RMT) can not only improve pulmonary function, but also simultaneously improve cardiovascular autonomic function and short-term functional outcomes in patients with PD. Pulmonary function was measured by the forced vital capacity (FVC), forced expiratory volume in one second (FEV1), maximum inspiratory pressures (MIP), and maximum expiratory pressures (MEP). Cardiovascular autonomic function was measured by the heart rate response to deep breathing (HRDB), Valsalva ratio, baroreflex sensitivity, and spectral analysis. The functional and severity scores were measured by the Hoehn and Yahr stage and Unified Parkinson’s Disease Rating Scale (UPDRS). These measures were evaluated in patients with PD before and after 3 months of RMT, compared with a control group of PD patients without RMT. The results showed significant improvement of clinical scores (total UPDRS and UPDRS I, II and III) after RMT (*p* < 0.0001). Concerning pulmonary function, the parameters of MIP and MEP improved significantly. The parameters of cardiovascular function also improved after RMT, although only HRDB reached statistical significance. Based on the results of our study, RMT can not only improve both pulmonary and cardiovascular autonomic function, but can also improve short-term functional outcomes in patients with PD.

## 1. Introduction

Parkinson’s disease (PD) is a progressive neurodegenerative disorder which manifests through a broad spectrum of motor and non-motor symptoms [1,2]. Patients with PD commonly have restrictive dysfunction with decreased respiratory muscle strength and endurance, and increased dyspnea is present in an estimated 50-60% of patients [3,4]. However, impaired pulmonary function usually goes unnoticed until in the advanced stages of the disease [3,4,5].

Autonomic disorders have been recognized as an important non-motor feature of PD [6,7]. It has been recognized that the autonomic nervous system plays a pivotal role in the regulation of lung ventilation, gas exchange, and airway smooth muscle function [8]. The observation that levodopa has a limited impact on the improvement of pulmonary function may suggest that respiration is not only associated with motor function, which is mediated mainly by a dopaminergic system, but also with other networks [9,10,11,12,13,14]. 

Breathing involves a complex interplay between the respiratory neuron network (RNN) and the central autonomic network (CAN) [15]. The RNN is involved in several key areas including motor cortex, somatosensory cortex, brainstem region including the medulla and pons, cerebellum, and insula, which were identified in a previous study [15]. In addition, a distributed network of forebrain regions can exert modulatory influences on the baroreflex and chemoreflex via their reciprocal projections within the brainstem, and exert modulatory influences on cardiovascular autonomic function [16]. These regions, including the hypothalamus, amygdala, anterior cingulate cortex, and insula, make up the CAN [17]. Both the RNN and CAN interact harmoniously to control respiratory muscle contractions, ensuring that normal blood gas levels are maintained during speech, volitional breathing, and ventilatory load. They can also exert modulatory influences on peripheral reflex activity via their reciprocal projections within the brainstem.

Previous studies on respiratory muscle training (RMT) for respiratory dysfunctions in patients with PD had relatively small patient numbers and less clinical grouping according to respiratory deficits. The outcome measurements were relatively limited, as they included only pulmonary and swallow safety or dysphagia-specific quality-of-life scales [18,19,20]. The results of these studies provide evidence that swallowing function or quality of life may be improved after expiratory muscle strength training. To date, there is a paucity of information addressing the relationship between respiratory and cardiovascular autonomic function in patients with PD. We propose the hypotheses that RMT for patients with PD can not only improve respiratory function, but also improve cardiovascular autonomic function and their functional outcomes as well. The successful translation of these approaches to treatment clinics offers the promise of improving the long-term outcome and the quality of life for PD patients.

## 2. Patients and Methods

### 2.1. Study Design and Participants

This single-center hospital-based prospective study enrolled 75 patients with PD, recruited consecutively from Chang Gung Memorial Hospital-Kaohsiung, a tertiary medical center and the main referral hospital serving a population of 3 million in southern Taiwan.

These patients had a definitive diagnosis of idiopathic PD according to the United Kingdom Parkinson’s Disease Society Brain Bank clinical diagnostic criteria [21] and attended follow-ups at the Neurology Outpatient Clinic for more than 6 months after titration of their daily anti-Parkinsonian agents to a steady dose in accordance with their clinical symptoms.

The exclusion criteria included: (1) newly diagnosed with PD or were on follow-up for less than 6 months, because their daily dose of anti-Parkinsonian agents was still under adjustment; (2) presence of focal neurological signs not related to the diagnostic criteria of PD; (3) active smoker or had quit smoking for less than 5 years; (4) had pulmonary diseases including chronic obstructive pulmonary diseases, bronchial asthma, or active pulmonary disease within 1 month of the study; (5) suffered from moderate-to-severe heart failure (NYHA class III and IV); and (6) had any type of arrhythmia that prevented BRS measurement, or pacemaker implantation due to any cause. For comparison, 37 age-, sex-, and body mass index (BMI)–matched patients with PD who were not willing to undergo RMT were included as controls. All the participants received verbal and written information about the purpose and process of our research, which was approved by the Chang Gung Memorial hospital’s institutional Review Committees on Human Research (IRB 201601640B0 and 201702037B0).

### 2.2. Respiratory Muscle Training (RMT)

The RMT protocol was performed in accordance with previous studies [22,23]. RMT was performed by an experienced respiratory therapist, using a Dofin Breathing Trainer. This is a hand-held pressure threshold device consisting of 10 training levels for inspiratory/expiratory muscle and swallow training (ALL-IN-ONE) with a color ball designed to indicate breathing status and strength. The device can be calibrated up to a pressure range of 5–39 cm H_2_O for inspiratory muscle training and 4–33 cm H_2_O for expiratory muscle training. RMT was applied to both generate the expiratory force for cough function and to strain inspiratory muscle for lung ventilation impairments. Patients with respiratory muscle weakness received inspiratory muscle training from 30% to 60% of maximum inspiratory pressures (MIP) through a respiratory trainer for two sets of 30 breaths or six sets of 10 repetitions. For patients with swallowing disturbance, expiratory muscle strengthening training commenced from 15% to 75% of threshold load of an individual’s maximum expiratory pressures (MEP), for five sets of 5 repetitions with one minute of rest between sets. The training resistance was adjusted accordingly, with one or two minutes of rest between sets. All patients were trained for 30 minutes twice per day, at least 5 days a week for 12 weeks, and were monitored by making a phone call to them once a week to check the compliance of RMT at home.

### 2.3. Clinical Assessment

All subjects underwent complete neurological pulmonary function, and cardiovascular autonomic function examinations upon enrollment and three months after starting RMT. All subjects underwent clinical assessment, pulmonary function testing, and autonomic studies during the “off” period of medication, which was defined as 8–12 hours after the latest dose of anti-parkinsonism agents. The clinical features recorded were age at disease onset (or age at the time of the first reported symptom attributable to the disease), sex, height, weight, body mass index, duration of disease (time from onset until follow-up), and education level. The daily dosage of anti-parkinsonism agents was converted into the levodopa-equivalent dose (LED) [24]. The severity or functional outcome of PD was assessed using Unified Parkinson’s Disease Rating Scale (UPDRS) and the Hoehn and Yahr (H-Y) stage [22,23]. The UPDRS total score was computed as the sum of subscores I, II, and III. For patients with fluctuation, assessment was administered in the “off period” to evaluate the possible influence of disease severity.

### 2.4. Testing Pulmonary Function

The pulmonary function of every patient with PD was evaluated by spirometry without exposure to bronchodilator, following the guidelines of the American Thoracic Society [25]. The forced vital capacity (FVC), forced expiratory volume in one second (FEV1), MIP, and MEP were recorded. The results of the pulmonary function tests are classified into three patterns as follows [26]-(i) obstructive pattern, which was defined as FEV1/FVC < 0.7; (ii) restrictive pattern, which was defined as FEV1/FVC *≥* 0.7 with FVC < 80%; and (iii) normal pattern, which was defined as FEV1/FVC *≥* 0.7 with FVC *≥* 80%.

### 2.5. Testing Autonomic Function

All of the patients with PD underwent a standardized evaluation of cardiovascular autonomic function, as described by Low [27]. The tests consisted of heart rate response to deep breathing (HRDB), Valsalva maneuver (VM), and 5 minutes of recording of resting blood pressure and heart rate for frequency-domain analysis and spontaneous baroreflex sensitivity (BRS). The tests were done between 9:00 am and 12:00 pm for all patients. No coffee, food, alcohol, or nicotine was permitted 4 hours before the tests. Patients on medications known to cause orthostatic hypotension or otherwise affect autonomic testing were asked to stop taking the drug for five half-lives, provided that it was not harmful to the patient’s well-being.

Heart rate was derived from continuously recorded standard three-lead ECGs (Ivy Biomedical, model 3000; Branford, CT, USA) while arterial blood pressure (BP) was continuously measured using beat-to-beat photoplethysmographic recordings (Finameter Pro, Ohmeda; Englewood, OH, USA). The following parameters were obtained through tests computed by Testworks (WR Medical Electronics Company, Stillwater, MN, USA)—HRDB, Valsalva ratio (VR), and BRS obtained by Valsalva maneuver (BRSVM). The detailed computing of HRDB and VR were as described by Low [27]. BRSVM was derived from changes in heart rate and BP during the early phase II of VM by applying least-squares regression analysis.

The spontaneous BRS by sequence method (BRSseq) [28] was computed using the Baroreflex Sensitivity Analysis software (Nevrokard, Slovenia). The program set the following criteria in computing BRSseq-1) systolic blood pressure (SBP) changes greater than 1 mmHg, 2) sequences longer than 3 beats, and 3) correlation coefficient greater than 0.85. Both bradycardiac (an increase in SBP that causes and increase in R-R interval [RRI]) and tachycardiac (a decrease in SBP that causes a decrease in RRI) sequences fulfilling the criteria were enrolled. The fluctuations of RRI and SBP were synchronous for some subjects, but a time-lag between these two fluctuations was present in other subjects. Therefore, BRSseq was calculated using the synchronous mode as well as the shift mode from 1 to 6 heart beat shifts for each subject [29]. The mode with the largest number of slopes was selected. The average slope of regression lines was taken as the measure of BRSseq.

Beat-to-beat RRI changes were interpolated using a third-order polynomial and were re-sampled with 0.5-sec intervals. The signals were then transformed to the frequency domain with fast Fourier transform using 512 samples. The spectral powers were divided into three frequency domains, high frequency (HF, 0.15–0.4 Hz), low frequency (LF, 0.04–0.15 Hz), and very low frequency (VLF, 0–0.04 Hz) [30]. The ratio between the powers of LF and HF (LF/HF ratio) was taken as an index of sympatho-vagal balance.

All of the above measures were evaluated twice for the enrolled patients, before and after 3 months of RMT, to compare with the control group of PD patients without RMT.

### 2.6. Statistical Analysis

Data are expressed as the mean ± standard deviation (SD) or median interquartile range (IQR). The categorical variables were compared using Chi-squared or Fisher’s exact tests. First, continuous variables that were not normally distributed were logarithmically transformed to improve normality, then compared between the two patient groups (RMT and control group) by using Student’s t-tests. Second, the changes of each parameter over three months were defined as the data at three-month follow-up minus the baseline data, and a correlation analysis was used to determine the relationship between the changes in MIP and MEP and the changes in parameters of cardiovascular functions and functional scores. Third, the changes between the baseline and three months post-RMT on the parameters of respiratory and cardiovascular autonomic functions and functional scores were compared using a paired t-test. Furthermore, a repeated-measures analysis of variance was used to compare the biomarkers and functional scores at enrollment and three months after RMT. Scheffe’s multiple comparison was used to analyze the intra-individual course of parameters over time and to compare the parameters of two different groups. All statistical analyses were conducted using SAS software version 9.1 (SAS Statistical Institute, Cary, NC, USA).

## 3. Results

### 3.1. Baseline Characteristics of the Patients

Table 1 shows the characteristics and baseline parameters of pulmonary and cardiovascular autonomic function between the study patients (*n* = 38) and the control group (*n* = 37). There was no significant difference between the two groups in terms of age, sex, BMI, duration of disease, LED, or disease severity scale. The baseline parameters of pulmonary and cardiovascular autonomic function were also similar in these two groups.

### 3.2. Correlation among Baseline Pulmonary Function, Cardiovascular Autonomic Function, and Disease Severity and Duration of PD

Table 2 shows the results of the correlation analysis between baseline pulmonary function and cardiovascular autonomic function, and between pulmonary function and disease severity or duration of PD. HRDB is significantly correlated with MEP and FVC, respectively. There was also a significant correlation between LF/HF ratio and FVC, i.e. the better the FVC, the higher the LF/HF ratio. There was a significant negative correlation between the H-Y stage and pulmonary function, including MIP, MEP, and FVC. In addition, MIP negatively correlated with the scores of UPDRS II, III, and total score, respectively.

### 3.3. Changes of Cardiovascular Autonomic Function, Pulmonary Function, and Functional Score in Study and Control Groups during the Study Period

The changes of cardiovascular autonomic function, pulmonary function, and functional scores during the study period are shown in Table 3. To exclude the possible effects of sex and age on the clinical scores, the hypothesis that the clinical scores were equal between sexes and ages was tested using ANCOVA. The parameters of cardiovascular function, including HRDB, VR, and BRS, all increased after training in the RMT group, although only the improvement of HRDB was statistically significant. Concerning pulmonary function, there was significant increase of MIP and MEP in the RMT group. The H-Y stage and UPDRS, including I, II, III, and total scores also improved significantly in the RMT group (*p* < 0.0001) (Figure 1). On the contrary, the data in the two measures were similar to the control group in functional scores, cardiovascular autonomic function, and pulmonary function. The only exception is that the parameter of FEV1 revealed a significant decrease in the second measurement. However, repeated measures ANOVA was done to compare the changes of two measures (baseline and 3-month follow-up) between these two groups but did not show a significant difference for each variable. The statistical analyses were as follows—HRDB (*p* = 0.656), VR (*p* = 0.824), BRSVM (*p* = 0.902), BRSseq (*p* = 0.724), LF/HF ratio (*p* = 0.425), MIP (*p* = 0.116), MEP (*p* = 0.096) and UPDRS, including I (*p* = 0.746), II (*p* = 0.591), III (*p* = 0.792), and total scores (*p* = 0.981).

### 3.4. The Amount of Change in Parameters of Cardiovascular Autonomic and Pulmonary Function in the RMT Group

Table 4 shows the results of the correlation analysis between the amount of change in the parameters of cardiovascular autonomic and pulmonary function, which was performed only in the RMT group. The amount of change was defined as the data in the three-month follow-up minus the baseline data. In VR, this change was significantly correlated with the changes of MIP and MEP, relatively. 

## 4. Discussion

To date, this is the first study to comprehensively evaluate the effects of RMT on both pulmonary and cardiovascular autonomic function in PD patients. This study also confirmed the hypothesis that RMT can improve not only pulmonary function but also cardiovascular autonomic function and the short-term functional outcomes in patients with PD.

The study examines the changes in pulmonary and cardiovascular autonomic function and clinical scores in PD patients before and after RMT. There are four major findings. First, there was significant correlation between disease duration/severity and pulmonary function (MIP/MEP or FVC) in PD patients. Second, after RMT, there were simultaneous improvements in function scores, pulmonary function, and cardiovascular autonomic function, although only the change in HRDB was statistically significant. Third, the changes in pulmonary function and in cardiovascular autonomic function have significant correlation. Fourth, the clinical scores (total UPDRS and UPDRS I, II and III) showed a significant improvement (*p* < 0.0001) after RMT therapy. 

The first finding is not surprising since it is consistent with the results of previous studies [3,4]. The motor impairment of PD patients is likely to result in restrictive pulmonary dysfunction. In addition, our recent study demonstrated that the abnormal pulmonary function group had smaller gray matter volume (GMV) in certain brain regions, and some of the parameters of pulmonary function were positively correlated with these regional GMV reductions. This finding suggests that central autonomic network and gray matter volume loss may underlie the respiratory dysfunction in PD patients [11].

Results showed that those PD patients receiving RMT had significant improvement not only in pulmonary function, but also in functional scores and cardiovascular autonomic function. The effects of RMT in patients with PD have been validated in several previous reports [18,19,20,31,32,33]. However, these studies only evaluated certain domains of function, such as respiratory or cough function, speech function (or phonation), swallowing function, or quality of life. The UPDRS was used in the current study because it is a multidimensional scale which evaluates PD patients on behavioral problems, motor function, daily activity function, and treatment complications. According to our results, there was a significant decrement of subscores I, II and III as well as the total score of UPDRS after RMT (Figure 1). There was no significant change in LED for these patients receiving RMT during the study period. Therefore, the improvement of these patients does seem to be caused by RMT rather than the medication effects. Although it seems intuitive for a patient to have better motor function once they achieve better pulmonary function, there may be a central mechanism explaining such change. Since the motor cortex is involved in RNN [15], RMT may have some activation effect on the motor cortex leading to the improvement of motor function. In addition, our study demonstrates that the improvement is not only in motor function, but also in function of daily activity and risk of treatment complications. Further study is necessary to explore the underlying neurophysiological basis of such findings.

The most important finding in the study is the improvement in cardiovascular autonomic function in patients with PD receiving RMT. To the best of our knowledge, it has not been reported before. The results showed that the autonomic parameters, including HRDB, VR, and BRS (both BRSVM and BRSseq), were increased after RMT, although only the increment of HRDB reached statistical significance. The improvement of HRDB and VR may be attributed to increased efforts of deep breathing and VM since the patients had gained strength in the respiratory muscles. Our data reveal that there was a significant correlation between the changes in VR and MIP/MEP. There is a connection or overlapping between CAN and RNN, according to previous reports. We suggest there may be central modulation causing the improvement of cardiovascular autonomic function after RMT, since the improvement of BRS is unlikely to be explained merely by the peripheral mechanism, i.e. increased respiratory effort. In addition, a recent study done in patients with Duchenne muscular dystrophy by Khokhar et al. also showed the parallel decline in pulmonary (MIP, MEP, and peak expiratory flow) and cardiovascular status [34]. Further physiological study is needed to elucidate the mechanism underlying this connection.

There were some limitations in this study. First, this is not a randomized blind-control study and we did not design any sham training. However, the neurologist who evaluated the patients’ functional scores was blind to the patients’ training status to avoid bias from the examiner. Second, there is a lack of image evidence of networks to further confirm the casual relationship. Finally, the study demonstrated only the short-term (3 months) effect of RMT. Detraining outcomes of expiratory muscle strength training have been reported [35]. More prospective multi-center investigations with long-term follow-ups are warranted to see how long the effects of RMT persist. 

## 5. Conclusions

Based on the results of our study, RMT can not only improve both pulmonary and cardiovascular autonomic function, but also improves the short-term functional outcomes in patients with PD.

## Figures and Tables

**Figure 1 jcm-09-00316-f001:**
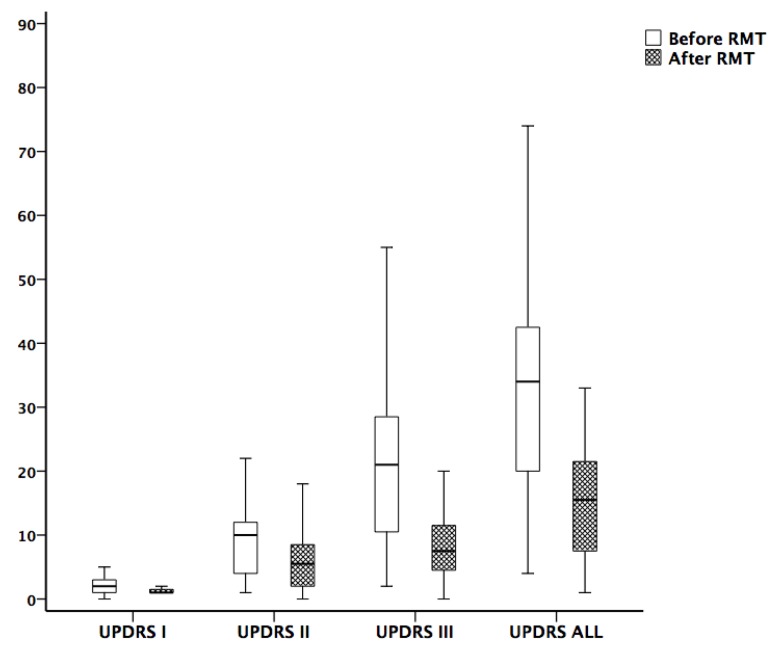
Comparison of Unified Parkinson’s Disease Rating Scale (UPDRS) before and after 3 months of respiratory muscle training. The improvement of UPDRS I, II, III, and total scores were significant.

**Table 1 jcm-09-00316-t001:** Baseline characteristics and parameters of pulmonary and autonomic function between groups of the study patients and diseases controls.

	*RMT Group (n = 38)*	*Disease Controls* *(n =* *37)*	*p Value*
Age, years	63.7 ± 10.0	64.5 ± 9.8	*0.757*
Sex (female; male)	21;17	12;12	*0.688*
*Body mass index (kg/m^2^)*	24.6 ± 4.2	23.8 ± 4.5	*0.496*
*Education, years*	9.1 ± 4.2	10.0 ± 5.4	*0.467*
*Disease duration, years*	5.5 ± 4.5	5.4 ± 4.3	*0.878*
*Levodopa equivalent dose (mg)*	731 ± 488	682 ± 379	*0.654*
**Disease severity scale**			
*Hoehn-Yahr stage*	2.0 [1.4, 2.5]	2.0 [1.0, 3.0]	*0.614*
*UPDRS* *^α^*	33.5 [16.5, 42.3]	33.0 [25.5, 43.5]	*0.817*
*UPDRS I* *^β^*	2.0 [1.0, 3.0]	1.5 [1.0, 3.0]	*0.511*
*UPDRS II* *^γ^*	10.0 [3.8, 12.0]	9.5 [7.3, 11.0]	*0.788*
*UPDRS III* *^δ^*	21.0 [9.8, 28.3]	20.5 [15.3, 29.3]	*0.756*
**Pulmonary function parameters**			
*FVC (% pred)*	85.5 ± 16.9	83.5 ± 15.5	*0.637*
*FEV1 (% pred)*	85.9 ± 13.7	85.2 ± 18.6	*0.891*
*FEV1/FVC*	80.7 ± 9.2	80.8 ± 9.2	*0.946*
*Maximum inspiratory pressures (MIP)*	80.8 ± 31.8	84.8 ± 40.6	*0.689*
*Maximum expiratory pressures (MEP)*	102.0 ± 32..6	90.0 ± 37.2	*0.209*
**Cardiovascular autonomic function**			
*Heart rate response to deep breathing (beats/min)*	7.3 ± 3.4	6.9 ± 3.7	*0.749*
*Valsalva ratio*	1.37 ± 0.17	1.30 ± 0.19	*0.290*
*BRSVM*	1.8 ± 0.9	1.7 ± 0.9	*0.835*
*BRSSeq*	7.1 ± 4.5	6.7 ± 3.6	*0.743*
*LF/HF ratio*	1.09[0.48, 1.75]	1.79[0.54, 2.37]	*0.236*

Values are expressed as mean ± SD or Median (interquartile range(IQR)). Abbreviations: Respiratory muscle training, RMT; FVC, Forced vital capacity; FEV1, forced expiratory volume in one second; BRSVM, baroreflex sensitivity obtained by Valsalva maneuver; BRSseq, baroreflex sensitivity obtained by sequence method; LF, low frequency; HF, high frequency; UPDRS, Unified Parkinson’s Disease Rating Scale; α = “Total UPDRS” score is the combined sum of parts I, II, and II. β = I. Mentation, behavior, and mood. γ = II. Activities of daily living (ADL). δ = III. Motor examination.

**Table 2 jcm-09-00316-t002:** Correlation among baseline pulmonary function, cardiovascular autonomic function, and disease severity and duration of PD.

	MIP		MEP		FVC	
Spearman Correlation	r	*p*	r	*p*	r	*p*
***Cardiovascular autonomic function***						
*HRDB (beats/min)*	*0.251*	*0.067*	*0.303*	***0.026 ****	*0.460*	***< 0.001 ******
*Valsalva ratio*	*0.082*	*0.571*	*0. 101*	*0.448*	*0.270*	*0.055*
*BRSVM (ms/mmHg)*	*−0.148*	*0.362*	*−0.032*	*0.843*	*−0.062*	*0.701*
*BRSseq (ms/mmHg)*	*0.129*	*0.377*	*0.168*	*0.249*	*0.059*	*0.682*
*LF/HF ratio*	*0.148*	*0.305*	*0.010*	*0.947*	*0.492*	***< 0.001 ******
*Disease duration, years*	*−0.254*	***0.046 ****	*−0.111*	*0.391*	*−0.156*	*0.222*
*Levodopa equivalent dose (mg)*	*−0.258*	***0.043 ****	*−0.108*	*0.402*	*−0.307*	*0.773*
***Disease severity scale***						
*Hoehn-Yahr stage*	*−0.298*	***0.020 ****	*−0.332*	***0.009 *****	*−0.374*	***0.003 ******
*UPDRS* *^α^*	*−0.333*	***0.008*****	*−0.201*	*0.117*	*−0.130*	*0.309*
*UPDRS I* *^β^*	*−0.159*	*0.216*	*−0.093*	*0.471*	*−0.134*	*0.296*
*UPDRS II* *^γ^*	*−0.327*	***0.009*****	*−0.171*	*0.183*	*−0.109*	*0.395*
*UPDRS III* *^δ^*	*−0.294*	***0.020****	*−0.207*	*0.107*	*−0.145*	*0.255*

Abbreviations: HRDB, heart rate response to deep breathing; BRSVM, baroreflex sensitivity obtained by Valsalva maneuver; BRSseq, baroreflex sensitivity obtained by sequence method; LF, low frequency; HF, high frequency; UPDRS, Unified Parkinson’s Disease Rating Scale; α = “Total UPDRS” score is the combined sum of parts I, II, and II. β = I. Mentation, behavior, and mood. γ = II. Activities of daily living (ADL). δ = III. Motor examination. * *p* < 0.05; ** *p* <0.01; *** *p* < 0.005.

**Table 3 jcm-09-00316-t003:** Changes of cardiovascular autonomic and pulmonary function and functional score between study and control groups during the study period.

	*RMT Group (n = 38)*		*Disease Control Group (n = 37)*	
	*Baseline*	*Follow-Up*	*Baseline*	*Follow-Up*
***Cardiovascular autonomic function***				
*HR_DB *	7.3 ± 3.4	9.1 ± 5.8 *	6.9 ± 3.7	7.7 ± 4.2
*Valsalva ratio*	1.37 ± 0.17	1.39 ± 0.24	1.30 ± 0.19	1.29 ± 0.16
*BRS_VM*	1.8 ± 0.9	2.1 ± 1.1	1.7 ± 0.9	1.7 ± 0.9
*BRS_Seq*	7.1 ± 4.5	7.4 ± 3.8	6.7 ± 3.6	7.2 ± 2.1
*LF/HF ratio*	1.09 [0.48, 1.75]	0.78 [0.44, 1.72]	1.79 [0.54, 2.37]	0.76 [0.39, 1.84]
***Pulmonary function parameters***				
*FVC (% pred)*	85.5 ± 16.9	81.3 ± 13.7	83.5 ± 15.5	83.0 ± 19.7
*FEV1 (% pred)*	85.9 ± 13.7	84.7 ± 16.0	85.2 ± 18.6	83.8 ± 18.4 *
*FEV1/FVC*	80.7 ± 9.2	82.1 ± 8.0	80.8 ± 9.2	80.7 ± 8.8
*Maximum inspiratory pressures (MIP)*	80.8 ± 31.8	103.5 ± 34.1 *	84.8 ± 40.6	99.0 ± 35.5
*Maximum expiratory pressures (MEP)*	102.0 ± 32..6	131.6 ± 34.8 *	90.0 ± 37.2	93.7 ± 43.9
***Disease severity score***				
*UPDRS I*	2.0 [1.0, 3.0]	1.0 [1.0, 1.5] *	1.5 [1.0, 3.0]	1.5 [1.0, 3.0]
*UPDRS II*	1.0 [3.8, 12.0]	5.0 [2.0, 8.5] *	9.5 [7.3, 11.0]	9.0 [8.0, 11.8]
*UPDRS III*	21.0 [9.8, 28.3]	7.0 [4.5, 11.5] *	20.5 [15.3, 29.3]	20.0 [16.3, 28.0]
*UPDRS ALL*	33.5 [16.5, 42.3]	15.0 [7.5, 21.5] *	33.0 [25.5, 43.5]	33.0 [27.3, 42.0]

Abbreviations: HRDB, heart rate response to deep breathing; BRSVM, baroreflex sensitivity obtained by Valsalva maneuver; BRSseq, baroreflex sensitivity obtained by sequence method; LF, low frequency; HF, high frequency; FVC, forced vital capacity; FEV1, forced expiratory volume in one second; MIP, maximum inspiratory pressure; MEP, maximum expiratory pressure; UPDRS, Unified Parkinson’s disease Rating Scale. The changes (baseline and 3-month follow-up) of cardiovascular autonomic function and pulmonary function parameters and disease severity score in different groups (RMT and disease control) were compared using paired-t test, respectively. * significant difference (*p* < 0.05) between follow-up and baseline.

**Table 4 jcm-09-00316-t004:** Correlation among the amount of change of pulmonary function, cardiovascular autonomic functions, and functional score in patients underwent ***respiratory muscle training*** during the study period.

	ΔMIP		ΔMEP	
Spearman Correlation	r	*p*	r	*p*
***Cardiovascular autonomic function***				
Δ*HRDB*	*0.141*	*0.412*	*0.039*	*0.822*
Δ*Valsalva Ratio*	*0.369*	*0.049 **	*0.383*	*0.040 **
Δ *BRSVM*	*−0.164*	*0.445*	*−0.083*	*0.701*
*ΔBRSseq*	*0.116*	*0.521*	*0.138*	*0.444*

Abbreviations: HRDB: heart rate response to deep breathing; BRSVM, baroreflex sensitivity obtained by Valsalva maneuver; BRSseq, baroreflex sensitivity obtained by sequence method; MIP, maximum inspiratory pressure; MEP, maximum expiratory pressure. Δ: Mean the changes during 3 months (Data in three-month follow-up minus baseline data). * *p* < 0.05.

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
