# Peer review of "Simultaneously Improved Pulmonary and Cardiovascular Autonomic Function and Short-Term Functional Outcomes in Patients with Parkinson’s Disease after Respiratory Muscle Training"

_jcm, 2020, doi:10.3390/jcm9020316_

Round 1

Reviewer 1 Report

Huang and colleagues present a prospective cohort study examining the effects of respiratory muscle training (RMT) on pulmonary function, cardiovascular autonomic function and functional outcomes in patients with idiopathic Parkinson's disease. They find significant improvement in all three categories. The study design and implementation are well done, although the authors recognize the limitations of a non-blinded and non-randomized study. The manuscript is also well written although I have some comments on the data analysis below.

Introduction:

Lines 70-72: what were the results of the previous referenced studies?

Patients and methods:

Lines 94-95: were controls matched to disease severity, baseline function, etc...?

Results:

Table 1: Please explain why the control group n is only 24 rather than 37 as stated in the methods section.

Table 2: small typo - HR_DB should be "beats"/min

Table 3: there is a difference between the n for the RMT group between Table 1 (n=38) and Table 3 (n=39)

Although you demonstrate a statistically significant change in HR_DB, MIP, MEP and UPDRS, there also appears to be a similar change in HR_DB, MIP and MEP in the control group. Although those changes were not statistically significant, this should be mentioned in your discussion and some statistical comparison between the changes in these two groups should be done. The change in UPDRS over the study period for the control group was not presented. Did this similarly improve in controls, but not reach statistical significance? Line 225 says "the data in the two measures were similar to the control group in functional scores, cardiovascular autonomic function and pulmonary function" but I'm not sure what that means exactly. If functional scores improved in both groups, it is harder to attribute the change to RMT.

Author Response

Lines 70-72: what were the results of the previous referenced studies?

Answer: The results of these studies provide evidence that swallowing function or quality of life may be improved after expiratory muscle strength training. The above description has been added to Lines 73-75.

Lines 94-95 (Lines 96-97 in rev_01): were controls matched to disease severity, baseline function, etc...?

Answer: As described in the RESULES section, 3.1: “There was no significant difference between the two groups in terms of age, sex, BMI, duration of disease, LED, or disease severity scale. The baseline parameters of pulmonary and cardiovascular autonomic function were also similar in these two groups.” (Lines 190-193)

Table 1: Please explain why the control group n is only 24 rather than 37 as stated in the methods section.

Answer: This single-center hospital-based prospective study enrolled 75 patients with PD, including RMT in 38 and the control group in the remaining 37. I am sorry for typing mistake. We correct the mistake.

Table 2: small typo - HR_DB should be "beats"/min

Answer: Exactly. It has been corrected. Thanks a lot.

Table 3: there is a difference between the n for the RMT group between Table 1 (n=38) and Table 3 (n=39)

Answer: It was a typing mistake in Table 3. It should be 38 rather than 39.

Reviewer 2 Report

The manuscript submitted for consideration by Chih-Cheng Huang et al., report the respiratory muscle training (RMT) improves pulmonary function as well as  cardiovascular autonomic function in patients with Parkinsons disease. Though the study is conducted in small group of patients, the results are well laid out and would be of critical interest. 

Author Response

Answer: Thanks for your comments.

Reviewer 3 Report

Important paper that reviewed the importance of RMT in short term improvement of strength of the respiratory pump in patients with Parkinson's disease. It also demonstrated the simultaneous improvement in pulmonary and cardiovascular capacity. No major concern. A very similar study was recently published in another neurological disease, duchenne muscular dystrophy which demonstrated the parallel decline in pulmonary (MIP, MEP, Peak expiratory flow) and cardiovascular status.

Khokhar, Arshjot, et al. "The Association between Pulmonary Function and Left Ventricular Volume and Function in Duchenne Muscular Dystrophy." Muscle & nerve (2019).

I recommend to include that paper as reference in introduction or discussion since this paper also cited the importance of simultaneous improvement of cardiac and pulmonary function in Parkinson's disease. 

Author Response

Answers: Thanks for your comments. The article has been included as the reference.
